# Electrolytes as predictors of fibro fatigue scores in Long-COVID patients

**Wasim Talib Mahdi Al Masoodi** [1,2]\*, **Sami Waheed Radhi**[3], **Hussein Kadhem Al-Hakeim**[4], **Habiba Khdair Abdalsada**[3]

**1** Department of Chemistry, College of Science, University of Kufa, Najaf, Iraq, **2** Faculty of Medicine, Department of Chemistry, University of Al-Ameed, Karbala, Iraq, **3** Faculty of Science, Department of Chemistry, University of Kufa, Najaf, Iraq, **4** College of Pharmacy, Al-Muthanna University, Samawah, Iraq

\* wasim.t@uokerbala.edu.iq

**Data Availability Statement:** The data used in this paper is available in a public repository figshare. com with the DOI https://doi.org/10.6084/m9. figshare.26654623.v1 at the link https://figshare.

## Abstract

### Background

The complex effects of Long-COVID, a syndrome marked by enduring symptoms after COVID-19 infection, with an emphasis on patients' differing degrees of fibro fatigue (FF). Electrolyte disturbances may affect the severity of FF and may be used as a predictive tool for severe FF in Long-COVID patients.

### Objective

The aim is to use the electrolyte levels for prediction of the Long-COVID patients with high FF levels.

### Methods

The electrolyte levels, calcium, and magnesium, as well as albumin and C-reactive protein levels were measured in 120 Long-COVID patients and 60 controls. FF scale was used for scoring the fatigue severity in all subjects. Patients were divided into high-FF (FF score>25) and moderate-FF group (FF score<25).

### Results

FF is the major effector on the serum biomarker levels. High-FF group had older people, longer disease durations, lower SpO2, higher CRP, and higher peak temperatures than the control group. High-FF group has a significant decrease in serum total and ionized calcium compared with the controls and low-FF group. After controlling the cofounders, the major factor controlling the levels of the measured biomarkers is the FF value (Partial $\eta^2$ = 0.468). The ROC-AUC analysis showed that the peak body temperature, Low-SpO2, high-CRP, and low electrolytes can predict the high-FF in a patient with Long-COVID with a moderate sensitivity and specificities (61.6–70%).

com/articles/dataset/Wasin-Al-Masoodi–Raw_
data_xlsx/26654623?file=48486631.

**Funding:** The author(s) received no specific
funding for this work.

**Competing interests:** The authors have declared
that no competing interests exist.

## Conclusion

Long-COVID patients have an elevation in FF score. The decline in electrolytes can predict the severity of FF with moderate sensitivities and specificities.

## Introduction

Long-COVID, which refers to the presence of persistent symptoms after an initial SARS-COV-2 infection, has become a complicated and difficult issue that impacts millions of people globally and lasts for a duration exceeding 12 weeks after the initial acute infection [1]. This can have long-term health and economic effects on the persons impacted and society [2]. Chronic sequelae in severe COVID-19 may be accompanied by tissue pathology (e.g., lung scarring, vascular thrombosis, post-intensive care psychological stress) that can be understood [3]. Anosmia and ageusia, which were frequent in pre-Omicron strains, are now rare for Long-COVID [4]. People with mild-to-moderate COVID-19 experience a wide range of mostly unexplained symptoms, including fatigue (sometimes with post-exertional malaise), neurocognitive dysfunction, "brain fog," sleep disturbances, and pain (myalgia, arthralgia, headache, chest pain) [3,5]. In COVID-19 survivors, fibromyalgia is common, and the risk is higher in women and those with dyspnea [6]. The primary symptom of fibromyalgia, known as myalgia, was observed in approximately 30% of individuals with post-acute sequelae of SARS-CoV-2 infection (PACS) [7]. Based on 41 publications, the worldwide incidence of COVID-19 is 43% [8]. However, certain studies have indicated a higher rate of 80% [9]. Many variables have been suggested as causes of fibromyalgia, including autoimmune illnesses, infections (especially viruses), and immunization [10–12]. There have been reports that COVID-19 infection makes fibromyalgia symptoms worse [13]. The fundamental symptoms of fibromyalgia, namely fatigue and myalgia, have been widely documented in patients after their COVID-19 infection [14]. Recent results reveal Omicron variations may reduce Long- COVID's prevalence [15]. Long-COVID can be found in people of all ages and levels of acute illness, but it is most common in Mild Acute Illness patients aged 36 to 50 who are not hospitalized [16]. Moreover, Long-COVID can significantly impact one's quality of life, functional capacity, and workability [17]. Long-COVID alters blood cell size and stiffness, reducing oxygen transport [18]. A significant worldwide health concern is the prevalence and consequences of Long-COVID, a chronic illness that develops after SARS-CoV-2 infection [19]. Irrespective of the severity of the original sickness, persons with Long-COVID suffer from persistent symptoms, with risk being influenced by characteristics such as age, sex, and immunization status [20–22]. Several studies link Long-COVID with symptoms like fibromyalgia and chronic fatigue syndrome, underscoring the importance of customized preventative and treatment approaches [23–25]. Furthermore, the fact that some symptoms, such as neurocognitive problems, are persistent, points to a complicated course that vaccinations may not be able to entirely treat [26]. The most frequently reported symptoms associated with Long-COVID are chronic fatigue, dyspnea, affective symptoms (anxiety and depression), and cognitive impairments [1,27–33]. The studies analyzed Long-COVID patients with varying levels of fibro fatigue (FF), revealing significant differences in age, illness duration, temperature, SpO2 levels, FF-Total scores, and CRP levels, with severe FF patients exhibiting advanced age, extended disease duration, elevated body temperature, and decreased oxygen saturation [34–38]. The present study used the electrolytes as predictors for differentiation between patients with severe and moderate fibro fatigue scores. This analysis helps us understand the detailed impact of this condition. This

research provides useful insights into the various features of Long-COVID, which can improve therapeutic decisions and specific interventions. In the next parts, the serum electrolytes in Long-COVID patients with moderate to high FF scores were examined and statistically analyzed for the prediction of the severity of FF in Long-COVID patients.

## Subjects and methods

### Subjects

This study employed case-control and retrospective cohort methodologies to compare individuals without Long-COVID to those with Long-COVID, and to evaluate the impact of acute-phase biomarkers on Long-COVID symptoms over the final three months of 2021. The study involved 120 people who had at least two symptoms of Long-COVID and had already been diagnosed with and treated for acute COVID-19 infection. The following are the criteria established by the World Health Organization (WHO) for post-COVID, which is also referred to as Long-COVID: (a) a verified infection with SARS-CoV-2; (b) symptoms that have continued beyond the initial phase of illness or emerged during the process of recovering from acute COVID-19 infection; (c) symptoms that have endured for a minimum of two months and are still present three to four months after the beginning of COVID-19; and (d) at least two symptoms that hinder daily functioning, such as fatigue, memory issues, and other symptoms [39]. During the initial stage of the infection, all patients were admitted to the Specific Quarantine Centers in Kerbala City including Al-Kafeel Super Specialty Hospital, Imam Al-Hassan Al-Mujtaba Teaching Hospital, Imam Al-Hussein Medical City of Kerbala, and Karbala Teaching Hospital for Children. SARS-CoV-2 and acute COVID-19 were diagnosed by senior physicians and virologists based on the following criteria: the occurrence of an acute respiratory syndrome accompanied by typical symptoms such as fever, coughing, and loss of taste and smell; a positive result from the rRT-PCR test; and a positive result from the SARS-CoV-2 IgM test. All patients have successfully recovered from the acute period, as indicated by negative rRT-PCR results. The control group consisted of 36 apparently healthy controls from the same geographical area and they have no significant difference with patients in age, BMI, and smoking ratio. These controls consisted of either staff members or individuals who were acquainted with them, such as family or friends. Additionally, we included individuals who tested negative for rRT-PCR and did not exhibit any symptoms of acute infection, such as a dry cough, sore throat, difficulty breathing, loss of appetite, flu-like symptoms, elevated body temperature, night sweats, or shivering. A senior psychiatrist reviewed clinical anamnesis and patient records to determine a person's career neuropsychiatric illness history. COVID-19 patients and controls excluded anyone with a history of psychiatric axis-1 disorders such as major depressive disorder, bipolar disorder, dysthymia, panic disorder, schizo-affective disorder, schizophrenia, psycho-organic syndrome, generalized anxiety disorder, or substance use disorders (other than tobacco use disorder). All participants were required to provide signed informed consent before taking part in the research. Approval for the research was granted by the University of Kufa's institutional ethics board (1657/2023) and the Kerbala Health Directorate: Training and Human Development Centre (Document No.0030/2023). Ethical and privacy standards set out by Iraqi and international bodies, such as the CIOMS Guideline, the Belmont Report, the International Conference on Harmonization of Good Clinical Practice, and the World Medical Association's Declaration of Helsinki, were adhered to throughout the research. The International Guidelines for the Safety of Human Research (ICH-GCP) are what our institutional review board follows.

## Clinical assessments

Senior psychiatrists conducted semi-structured interviews with both control subjects and Long-COVID patients 3–4 months after infection to gather socio-demographic and clinical information. The senior psychiatrist utilized the 12-item FF scale to examine the severity of chronic tiredness and fibromyalgia symptoms in individuals who had previously had acute COVID-19. The patient group was classified according to the FF score into two groups; severe fatigue group (FF>25) and moderate fatigue group (FF<25). The evaluation took place three to four months after the initial infection [40]. It would be unclear to use the total scores of all three scales to evaluate affective symptoms because they take into account both physical and mental (affective) problems. The diagnosis of TUD was made using the DSM-5 criteria. The BMI was determined by dividing the weight of the body in kilograms by the height in meters squared.

## Assays

Between 7:30 and 9:00 a.m., before participants ate breakfast, researchers took a 5 mL venous blood sample and deposited it into sterile containers to use in the study. The investigation excluded hemolyzed materials following the centrifugation of clotted blood samples at a speed of 1006 Xg for 5 minutes. The resulting serum was then separated and transferred into three new Eppendorf tubes for further testing. Estimation of serum CRP, Albumin, T.Ca, I.Ca, T. Mg, I. Mg, T.Ca/Mg, I.Ca/Mg were conducted using kits provided by Agappe® Diagnostics Ltd., (Cham, Switzerland). Laboratory workers were unaware of clinical data. The study examined patient records to determine the lowest levels of SpO2 and PBT readings during the acute phase. The study kept track of which vaccines were given to each subject, which were made by AstraZeneca®, Pfizer®, or Sinopharm®.

## Biostatistical analysis

Analyzing variance and contingency tables test determined scale variable differences and nominal variable associations across groups. The correlations between SpO2, PBT, clinical evaluations, and biomarkers were examined using Pearson's product-moment correlation coefficients. A univariate general linear model (GLM) analysis was employed to examine the correlations between categorization and biomarkers, while taking into account the effects of TUD, sex, age, BMI, and education. The between-subjects effects analysis was utilized for Long-COVID patients with high FF significant biomarker changes. The primary study examined biomarker-clinical data relationships in a research sample of patients and controls without constraints. Within the subset of Long-COVID patients, we sought for correlations as part of the secondary analysis. The threshold for statistical significance was established as a p-value of 0.05 for two-tailed tests. An analysis of variance with three groups and up to five variables should yield 151 participants (calculated using GPower 3.1.9.7). Consequently, we registered a total of 156 people, with 36 being controls and 120 being Long-COVID patients.

## Results

### Sociodemographic, clinical, and biochemical data in the study groups

Table 1 shows the sociodemographic and clinical characteristics of Long-COVID patients and control group with varying FF levels.

Three groups of 30 people were studied: a control group (Group A), Long-COVID patients with FF scores below 25 (Group B), and those with FF scores over 25 (Group C). Patients with FF scores above 25 were older than those in the other groups. The groups did not differ in BMI

**Table 1. Sociodemographic, clinical, and biochemical data in Long-COVID patients with severe and moderate fibro fatigue (FF) scores as well as the control group.**

| Parameter | Control [A] n = 30 | Long-COVID FF<25 [B] n = 29 | Long-COVID FF>25 [C] n = 31 | F/χ² | df | p |
|---|---|---|---|---|---|---|
| Age        Years | 32.87±7.01 [B] | 39.31±10.12 [A] | 35.97±9.74 | 3.733 | 2/87 | 0.028 |
| BMI        kg/m² | 27.16±2.83 | 27.51±4.13 | 27.39±4.63 | 0.060 | 2/87 | 0.942 |
| Sex        M/F | 8/22 | 12/17 | 15/16 | 3.138 | 2 | 0.208 |
| Single/Married | 5/25 | 4/25 | 3/28 | 0.652 | 2 | 0.722 |
| Rural/Urban | 5/25 | 7/22 | 6/25 | 0.527 | 2 | 0.768 |
| Employment No/Yes | 5/25 | 6/23 | 7/24 | 0.346 | 2 | 0.841 |
| Education        Year | 12.97±4.88 | 11.17±5.54 | 11.87±4.45 | 0.982 | 2/87 | 0.379 |
| Smoking        No/Yes | 18/12 | 18/11 | 20/11 | 0.133 | 2 | 0.936 |
| Vaccination A,PF,S | 11/14/5 | 16/9/4 | 12/15/4 | 9.388 | 2 | 0.153 |
| Duration of Dis. Day | 8.63±5.74 [B,C] | 14.34±8.97 [A,C] | 16.16±9.19 [A,B] | 7.073 | 2/87 | 0.001 |
| Period of Cure Months | 16.07±5.85 | 18.38±6.48 | 16.26±5.69 | 1.345 | 2/87 | 0.266 |
| Highest temp.˚C | 38.50±0.79 [C] | 38.69±0.83 [C] | 39.15±0.92 [A,B] | 4.676 | 2/87 | 0.012 |
| SpO2        % | 92.97±3.48 [B,C] | 86.93±6.29 [A,C] | 74.97±14.74 [A,B] | 27.969 | 2/87 | <0.001 |
| FF-TOTAL | 6.07±2.15 [B,C] | 32.55±9.89 [A,C] | 35.97±8.69 [A,B] | 136.613 | 2/87 | <0.001 |
| CRP (Quantitative) mg/l | 3.50±1.35 [B,C] | 5.35±3.56 [A,C] | 7.81±5.09 [A,B] | 10.483 | 2/87 | <0.001 |
| CRP (Qualitative) -Ve/+Ve | 28/2 [B,C] | 17/12 [A] | 17/14 [A] | 12.646 | 2 | 0.002 |
| Albumin        g/l | 4.56±0.19 | 4.66±0.25 | 4.60±0.41 | 0.817 | 2/87 | 0.445 |
| T.Ca        mM | 2.34±0.24 [C] | 2.32±0.34 [C] | 2.16±0.13 [A,B] | 4.787 | 2/87 | 0.011 |
| I.Ca        mM | 1.30±0.07 [C] | 1.30±0.08 [C] | 1.25±0.04 [A,B] | 5.196 | 2/87 | 0.007 |
| T.Mg        mM | 1.06±0.35 | 1.03±0.11 | 0.93±0.09 | 3.063 | 2/87 | 0.052 |
| I. Mg        mM | 0.74±0.23 | 0.72±0.07 | 0.65±0.06 | 3.063 | 2/87 | 0.052 |
| T.Ca/Mg | 2.54±1.19 | 2.29±0.29 | 2.18±0.41 | 0.904 | 2/87 | 0.409 |
| I.Ca/Mg | 1.99±0.84 | 1.82±0.16 | 1.94±0.18 | 0.919 | 2/87 | 0.403 |

[A,B,C]: Pairwise comparison, BMI: body mass index, Highest temp.: The highest temperature recorded during the acute infection, SpO2: Saturated oxygen percentage, FF-TOTAL: fibro fatigue total score, T.Mg: Total magnesium, I.Mg: ionized magnesium, T.Ca: total calcium, I.Ca: ionized calcium, I.Ca/Mg: ionized calcium-to-magnesium ratio, CRP: C-reactive protein, and T.Ca/Mg: total calcium-to-magnesium ratio, and Vaccination (A, PF, S): vaccination with AstraZeneca, Pfizer, or Sinopharm.

or education. Patients' groups had longer disease durations and lower oxygen saturation (SpO2) than the control group. Significant differences in the peak body temperatures were seen among the groups, with Group C having higher temperatures. There are no significant variations in albumin levels across groups. Severe FF patients had the lowest SpO2 and higher FF-TOTAL score and CRP than other groups. Group C patients have significant decrease in serum total and ionized calcium compared with Groups A and B. While there is no significant difference in the serum total and ionized magnesium, as well as the calcium to magnesium ratios.

## Results of the multivariate generalized linear model (GLM)

The multivariate generalized linear model (GLM) analysis in Table 2 examined the biomarker effects of Long-COVID with high FF score (>25) between patients.

The study found significant differences in all parameters for Long-COVID patients with high FF (Partial $\eta^2$ = 0.468). Period of cure, disease duration, sex, age, BMI, smoking, occupation, and marital status had no significant influence. The between-subjects effects analysis showed substantial biomarker changes in Long-COVID patients with high FF. Specifically, SpO2 (Partial $\eta^2$ = 0.238) showed a significant effect, suggesting that the high FF group had a

**Table 2. Results of the multivariate generalized linear model (GLM) analysis and the between-subjects effects of the Long-COVID with high fibro fatigue (FF) on the biomarkers.**

| Test | Dependent Variable | Effect | F | p | Partial $\eta^2$ |
|---|---|---|---|---|---|
| Multivariate Tests | All measured parameters | High FF>25 Long-COVID | 4.108 | 0.001 | 0.468 |
| | | Period of Cure | 1.313 | 0.259 | 0.220 |
| | | Duration of Disease | 0.992 | 0.461 | 0.175 |
| | | Sex | 0.857 | 0.570 | 0.155 |
| | | Age | 0.841 | 0.583 | 0.153 |
| | | BMI | 0.838 | 0.586 | 0.152 |
| | | Smoking | 0.734 | 0.675 | 0.136 |
| | | Employment | 0.630 | 0.765 | 0.119 |
| | | Marital status | 0.566 | 0.817 | 0.108 |
| Tests of Between-Subjects Effects | High FF>25 Long-COVID | SpO2 | 15.580 | <0.001 | 0.238 |
| | | CRP | 5.743 | 0.020 | 0.103 |
| | | Highest temp. | 4.900 | 0.031 | 0.089 |
| | | T.Ca/Mg | 3.737 | 0.059 | 0.070 |
| | | I.Ca | 3.540 | 0.066 | 0.066 |
| | | T.Ca | 3.020 | 0.088 | 0.057 |
| | | I.Ca/Mg | 2.576 | 0.115 | 0.049 |
| | | I. & T.Mg | 1.836 | 0.181 | 0.035 |
| | | Albumin | 0.366 | 0.548 | 0.007 |

SpO2: oxygen saturation percentage, BMI: body mass index, T.Mg: Total magnesium, I.Mg: ionized magnesium, T.Ca: total calcium, I.Ca: ionized calcium, I.Ca/Mg: ionized calcium-to-magnesium ratio, CRP: C-reactive protein, and T.Ca/Mg: total calcium-to-magnesium ratio.

reduced oxygen saturation. Furthermore, different levels of impact were shown by CRP (Partial $\eta^2 = 0.103$), followed by the highest temperature during the acute phase of infection (Partial $\eta^2 = 0.089$). Other biomarkers are not affected by the FF score ($p > 0.05$) after controlling the cofounders.

## Analysis of receiver operating characteristic-area under curve (ROC-AUC)

1. ROC-AUC analysis for prediction of high FF in patients using clinical data

The *ROC-AUC analysis* to identify fatigue in Long-COVID patients based on clinical data collected during the acute phase of the SARS-CoV-2 infection is shown in Table 3 and Fig 1.

Remarkably, the maximum temperature observed during the infection, using a threshold of 39.2˚C, showed the patients would have a higher FF score with a sensitivity of 70% and sensitivity of 69%. Additionally, the decline of the SpO2 was lower than the cut-off level of 82.5% indicating a good ability to predict the high FF in a patient with a sensitivity of 65.5% and

**Table 3. Receiver operating characteristic-area under curve (AUC) analysis of the clinical data during the acute SARS-COV-2 infection for predicting fatigue in Long-COVID patients.** CI: Confidence interval.

| Test | Cut-off | Sensitivity % | Specificity % | Youden's J statistic | AUC (SE) | 95% CI of AUC | p-value |
|---|---|---|---|---|---|---|---|
| Highest temp.˚C | 39.2 | 70.0 | 69.0 | 0.39 | 0.66(0.07) | 0.52–0.80 | 0.033 |
| SpO2% * | 82.5 | 65.5 | 61.6 | 0.27 | 0.75(0.07) | 0.61–0.85 | 0.002 |
| Duration of Dis. (Days) | 13.5 | 52.2 | 49.3 | 0.15 | 0.56(0.08) | 0.42–0.71 | 0.403 |
| Period of Cure (month) * | 14.5 | 51.6 | 48.9 | 0.05 | 0.60(0.07) | 0.45–0.74 | 0.201 |

(*): The decrease in these parameters can predict the patients with severe FF scores, SpO2: oxygen saturation percentage.

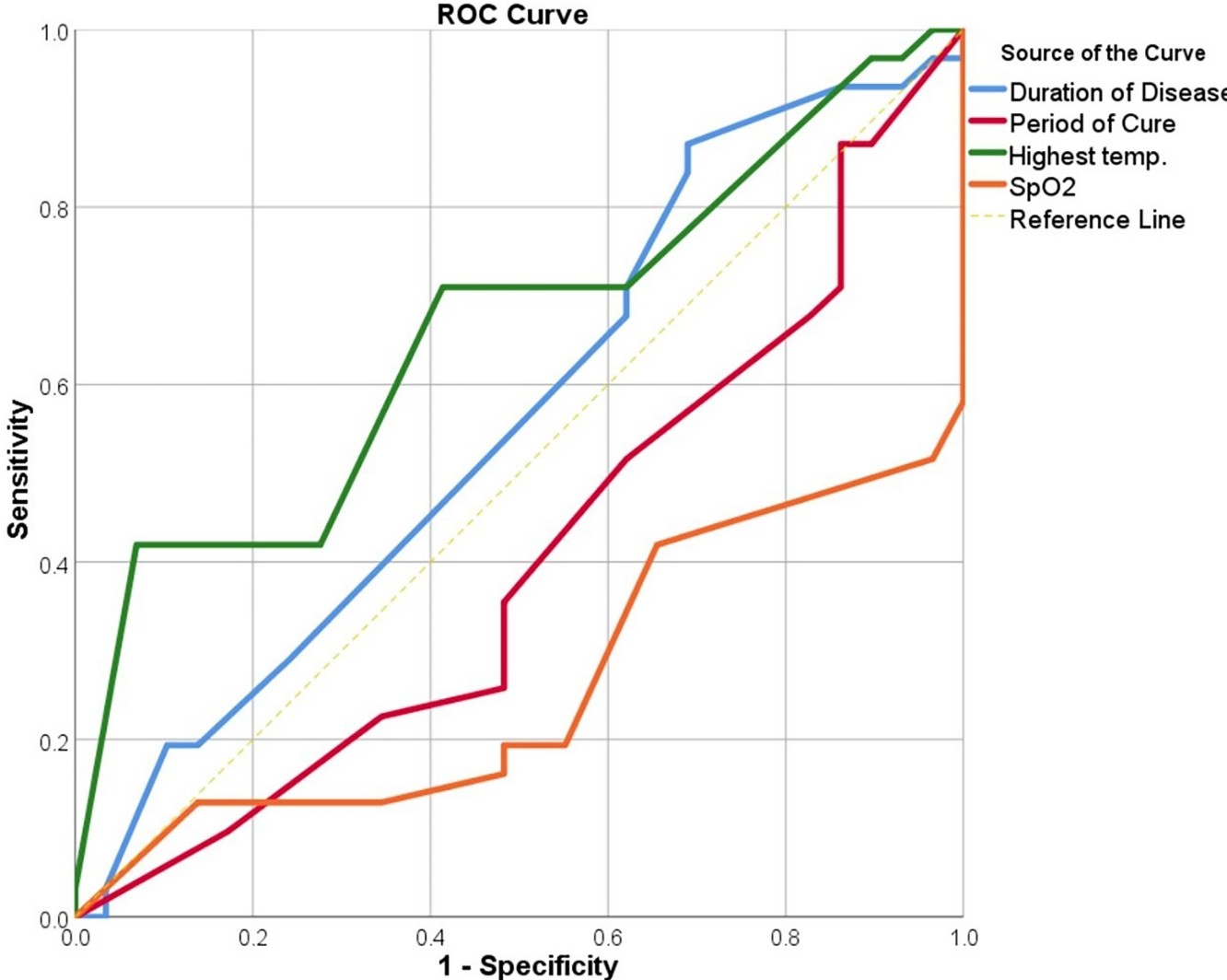

**Fig 1. Receiver operating characteristic curves of the clinical data during the acute SARS-COV-2 infection for predicting fatigue in Long-COVID patients.** SPO2: Oxygen saturation percentage.

specificity of 61.6%. However, disease duration and cure time did not predict the severity of FF scores.

2. ROC-AUC analysis for prediction of high FF in patients using serum biomarkers

Long-COVID patients with severe and moderate FF scores are distinguished by several serum biomarkers as seen in Fig 2 and Table 4.

The decrease in total magnesium at a cut-off value <0.98 mM, and ionized magnesium at a cut-off value<0.68 mM can predict the severity of FF with a sensitivity = 66%, and specificity = 66%. Also, the decrease in albumin at a cut-off value <46.4 g/l can predict the severity of FF with a sensitivity = 62%, and specificity = 66%. The increase in total calcium at a cut-off value <2.19 mM can predict the severity of FF with a sensitivity = 64%, and specificity = 66%. Ionized calcium at a cut-off value<1.26 mM can predict the severity of FF with a sensitivity = 66%, and specificity = 64%. Serum CRP higher than the cut-off value of 5.31 mg/l can predict the high FF with a sensitivity = 58%, and specificity = 55%. Also, the increase in I.Ca/Mg

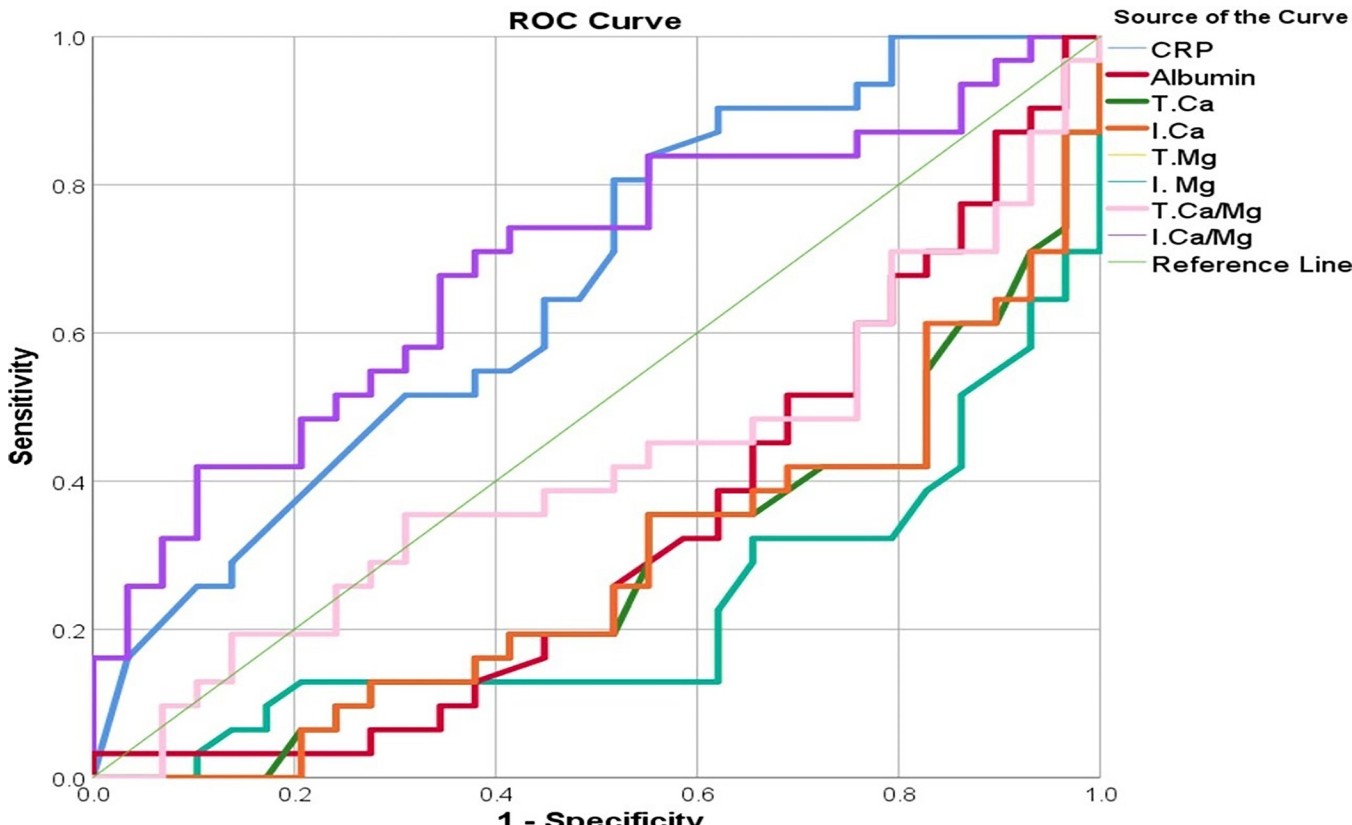

**Fig 2. Receiver operating characteristic curves of measured parameters in differentiating between Long-COVID patients with severe and moderate FF scores.** Footnote: CRP: C-reactive protein, T.Ca, and T.Mg: total calcium and magnesium, I.Ca and I.Mg: ionized calcium and magnesium.

ratio higher than the cut-off value of 1.87 can predict a high FF with a sensitivity and specificity of 65%. According to the results, T.Ca/Mg is not a good predictor of FF scores ($p > 0.05$).

## Discussion

The results in Table 1 revealed that the Long-COVID symptoms are related to the clinical state of patients (SPO2, body temperature, and duration of illness) during the acute phase of the

**Table 4. Receiver operating characteristic-area under curve (AUC) analysis of measured parameters in Long COVID group differentiating between patients with severe and moderate FF score.** CI: Confidence interval.

| Test | Cut-off | Sensitivity % | Specificity % | Youden's J statistic | AUC (SE) | 95% CI of AUC | p |
|---|---|---|---|---|---|---|---|
| T.Mg mM* | 0.98 | 0.66 | 0.68 | 0.34 | 0.77(0.06) | 0.65–0.89 | <0.001 |
| I. Mg mM* | 0.68 | 0.66 | 0.68 | 0.34 | 0.77(0.06) | 0.65–0.89 | <0.001 |
| T.Ca mM* | 2.19 | 0.64 | 0.66 | 0.30 | 0.72(0.06) | 0.59–0.85 | 0.003 |
| I.Ca mM* | 1.26 | 0.66 | 0.64 | 0.30 | 0.72(0.06) | 0.59–0.85 | 0.003 |
| I.Ca/Mg | 1.87 | 0.65 | 0.65 | 0.30 | 0.69(0.07) | 0.56–0.83 | 0.011 |
| Albumin g/l* | 46.40 | 0.62 | 0.66 | 0.28 | 0.67(0.07) | 0.53–0.81 | 0.021 |
| CRP mg/l | 5.31 | 0.58 | 0.55 | 0.13 | 0.67(0.07) | 0.53–0.80 | 0.028 |
| T.Ca/Mg # | 2.35 | 0.45 | 0.45 | -0.10 | 0.58(0.08) | 0.43–0.73 | 0.277 |

(*): The decrease in these parameters is predictive of severe FF scores. T.Mg: Total magnesium, I.Mg: ionized magnesium, T.Ca: total calcium, I.Ca: ionized calcium, I. Ca/Mg: ionized calcium-to-magnesium ratio, CRP: C-reactive protein, and T.Ca/Mg: total calcium-to-magnesium ratio.

infection with SARS-COV-2. These higher body temperatures for a longer time with severe hypoxia may cause damage to the microvessels in multiple organs including the brain and lung [41]. There is also a risk of persistent microcoagulation caused by the illness [42]. These factors can explain the symptoms of fibrofatigue scores in our patient groups. Age, disease duration, temperature, SpO2, FF-TOTAL scores, and CRP had significant differences across groups [36,43,44]. More severe FF patients were older and had longer disease durations, higher temperatures, poorer oxygen saturation, and higher FF-Total and CRP scores. These findings shed light on how FF may affect Long-COVID patients' clinical and biochemical profiles, helping to comprehend their health state [45]. The study's results contribute to the ongoing discussion on the relationship between Long-COVID and fibromyalgia, offering potential insights into shared symptoms and underlying mechanisms. There is previous evidence that viral infections are related to the development of fatigue, severe depression, and anxiety [46–48]. The increase in FF score in Long-COVID patients has been reported recently [49,50]. Within six months following the onset of the first COVID-19 symptom, about one-third of COVID-19 survivors experience neuropsychiatric symptoms, such as sleeplessness, anxiety, or depression [51].

The multivariate GLM study, in Table 2, showed that the presence of high FF in Long-COVID patients is the main significant factor affecting the variations in the sera level of the measured biomarkers. Other cofounders (period of cure, duration of disease, sex, age, BMI, smoking, employment, and marital status) have no significant effect size ($p < 0.05$) on the level of the measured biomarkers. These results are important as they indicate a variation in the blood component that affects the overall state of energy of the Long-COVID patients. Previous studies support our explanation by showing that Long-COVID and fibromyalgia may have some of the same symptoms [43,52,53]. The findings add to the Long-COVID fibromyalgia debate by shedding light on overlapping symptoms and processes. The strong influence of FF on Long-COVID patients' clinical and biochemical profiles highlights the need for more research to distinguish the two illnesses and provide appropriate care [36,54]. Long-COVID patients with high FF had significant biomarker changes in the between-subjects effects analysis, indicating altered clinical and biochemical profiles. Long-COVID patients with high FF had lower SpO2 and variable effects on inflammatory markers, temperature, and metabolic measures. These data suggest that changed biomarkers, particularly oxygen saturation and inflammatory markers, may affect Long-COVID patients' health outcomes [43,53–55]. By providing possible insights into common symptoms and underlying mechanisms, the study's findings add to the continuing conversation about the connection between Long-COVID and fibromyalgia [52].

The ROC analysis, in Table 3, shows that acute SARS-CoV-2 clinical data especially the peak body temperature and SpO2 can predict fatigue in Long-COVID patients. Importantly, the highest temperature recorded during the infection and the level of SpO2 demonstrated a moderate to high level of accuracy in predicting fatigue in patients with Long-COVID. More precisely, a maximum temperature limit of 39.2˚C demonstrated a modest degree of predictive precision, whereas a SpO2 threshold of 82.5% suggested a strong capability to predict fatigue in Long-COVID patients. These findings suggest that specific clinical parameters during the acute phase of SARS-CoV-2 infection, such as temperature and SpO2, may serve as early predictors of fatigue in Long-COVID patients, enabling focused treatment and management strategies [43,56,57].

According to the second ROC analysis in Table 4 and Fig 2, using electrolyte levels as predictive tools, the serum level of T.Mg, I.Mg, T.Ca, I.Ca, and albumin moderately predicted the severity of FF scores in Long-COVID patients. While CRP and T.Ca/Mg and T.Ca/Mg ratios have no significant predictability of FFscores in the patients ($p < 0.005$). These findings indicated the importance of electrolytes in the immunity and muscle functions that in turn affect

the severity of the fatigue symptoms. Previous research has addressed the connection between Long-COVID and fibromyalgia [36,43,58]. In this study, the electrolytes are other factors that may be useful in predicting FF severity in Long-COVID patients. The study's findings align with prior research that has established symptom severity scores as a prognostic indicator for Long-COVID [57]. The findings reveal probable underlying processes between Long-COVID and fibromyalgia, given common symptoms and mechanistic roles [36,43,58]. The results emphasize the importance of these findings in clinical treatment decision-making [25,43,52,56,57]. The study's findings may shed light on Long-COVID and fibromyalgia's common symptoms and causes. Correction of the alteration in the magnesium and calcium in Long-COVID patients may be necessary to monitor their effects on the FF score in those patients. These results require confirmation, further future work is necessary to investigate the possibility of the effect of other measurable parameters on the FF state in Long-COVID patients.

## Limitations

Due to its case-control design, the study cannot establish causality. Self-reported data may create recollection biases, and double stratification may reduce statistical power. Although the study had a large sample size, the limited number of measured biomarkers may limit its application. Prospective studies are needed to validate clinical parameter prediction since retrospective analysis creates biases. Research on FF in Long-COVID patients must address these constraints.

## Conclusions

This study shows that age, disease duration, temperature, oxygen saturation, and important biochemical indicators affect FF in Long-COVID patients. The observed differences in biochemical and clinical profiles highlight the widespread impact of FF on Long-COVID people's health outcomes. The high predictability of T.Mg, I.Mg, T.Ca, I.Ca, and Albumin in severe FF patients underscores their potential utility in distinguishing fatigue severity. A lot of important information was learned in this study that can help doctors treat FF in Long-COVID patients.

## Acknowledgments

The author would like to thank the senior pulmonologists Ammar Abbas Neamh and Maytham Abdulameer Al Maamory.

The authors express gratitude to various hospitals in Iraq for their assistance in gathering sample material, senior pulmonologists Maytham Abdulameer Al Maamory and Ammar Abbas Neamh, and internal lab staff for estimating biomarker levels.

## Author Contributions

**Conceptualization:** Wasim Talib Mahdi Al Masoodi.

**Data curation:** Wasim Talib Mahdi Al Masoodi.

**Formal analysis:** Wasim Talib Mahdi Al Masoodi.

**Funding acquisition:** Wasim Talib Mahdi Al Masoodi.

**Investigation:** Wasim Talib Mahdi Al Masoodi.

**Methodology:** Wasim Talib Mahdi Al Masoodi, Sami Waheed Radhi, Hussein Kadhem Al-Hakeim, Habiba Khdair Abdalsada.

**Project administration:** Wasim Talib Mahdi Al Masoodi, Sami Waheed Radhi, Habiba Khdair Abdalsada.

**Resources:** Wasim Talib Mahdi Al Masoodi.

**Software:** Wasim Talib Mahdi Al Masoodi, Hussein Kadhem Al-Hakeim.

**Supervision:** Wasim Talib Mahdi Al Masoodi, Sami Waheed Radhi, Hussein Kadhem Al-Hakeim, Habiba Khdair Abdalsada.

**Validation:** Wasim Talib Mahdi Al Masoodi, Sami Waheed Radhi.

**Visualization:** Wasim Talib Mahdi Al Masoodi, Sami Waheed Radhi, Hussein Kadhem Al-Hakeim, Habiba Khdair Abdalsada.

**Writing – original draft:** Wasim Talib Mahdi Al Masoodi.

**Writing – review & editing:** Hussein Kadhem Al-Hakeim.

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
