## [Decision Letter · Decision Letter 0]

11 Jul 2024

PONE-D-24-09563Electrolytes as predictors of Fibro Fatigue (FF) scores in Long COVID patients: A focus on Calcium and MagnesiumPLOS ONE

Dear Dr. Al-Masoodi,

Thank you for submitting your manuscript to PLOS ONE. After careful consideration, we feel that it has merit but does not fully meet PLOS ONE’s publication criteria as it currently stands. Therefore, we invite you to submit a revised version of the manuscript that addresses the points raised during the review process.

**ACADEMIC EDITOR:**You have to be consistence to say COVID-19 because I saw some COVID-19 and some COVIDYou need to revise the margin type of referencesYou need to explain 1 or 2 sentences about the table and figure.==============================

We look forward to receiving your revised manuscript.

Kind regards,

Diana Laila Ramatillah, PhD

Academic Editor

PLOS ONE

Journal Requirements:

5. Please ensure that you include a title page within your main document. You should list all authors and all affiliations as per our author instructions and clearly indicate the corresponding author.

6. Please include your tables as part of your main manuscript and remove the individual files. Please note that supplementary tables (should remain/ be uploaded) as separate ""supporting information"" files

**Additional Editor Comments:**

Electrolytes as predictors of Fibro Fatigue (FF) scores in Long COVID patients: A focus on Calcium and Magnesium

1. You have to be consistence to say COVID-19 because I saw some COVID-19 and some COVID

2. You need to revise the margin type of references

3. You need to explain 1 or 2 sentences about the table and figure.

Reviewers' comments:

Reviewer's Responses to Questions

**Comments to the Author**

1. Is the manuscript technically sound, and do the data support the conclusions?

Reviewer #1: Yes

Reviewer #3: Yes

2. Has the statistical analysis been performed appropriately and rigorously? 

Reviewer #1: Yes

Reviewer #3: Yes

3. Have the authors made all data underlying the findings in their manuscript fully available?

Reviewer #1: Yes

Reviewer #3: Yes

4. Is the manuscript presented in an intelligible fashion and written in standard English?

Reviewer #1: Yes

Reviewer #3: Yes

5. Review Comments to the Author

Reviewer #1: The presented topic is significant, and the authors used electrolyte levels to predict long-term COVID-19 patients with high FF levels and discussed the observed differences in biochemical and clinical profiles highlight the widespread impact of FF on Long-COVID people's health outcomes. They have proven their claims through the results provided with statistical operations and performed the ROC-AUC analysis However, I have one concern that the authors have to consider and resolve, as follows:

1. At the end of the introduction part, there is no general overview of the arrangement of the rest of the manuscript; it should provide an overview of the following sections of the manuscript.

Reviewer #3: Q- Were there any notable biases or limitations in the selection of participants, such as age distribution or comorbidities?

Q- Were there any control measures taken to minimize confounding factors that could influence the results?

Q- What are the potential clinical implications of the study findings for managing Long-COVID patients with fibrofatigue?

Q- How might the predictive biomarkers identified in this study impact clinical practice or patient care strategies?

Q- Are there any recommendations for further research or validation studies based on the findings?

6. PLOS authors have the option to publish the peer review history of their article (what does this mean?). If published, this will include your full peer review and any attached files.

Reviewer #1: No

Reviewer #3: **Yes: **Dr. Muhammad Suleman

---

## [Author Response · Author response to Decision Letter 0]

24 Jul 2024

Response to reviewers

PONE-D-24-09563

Electrolytes as predictors of Fibro Fatigue (FF) scores in Long COVID patients: A focus on Calcium and Magnesium

PLOS ONE

Dear Editor, 

 Thank you very much for your effort and to the reviewers for reviewing our manuscript and adding important notes and requirements. We responded positively to all the editor's and reviewers' concerns. We highlighted the amendments in the revised manuscript along with our response to reviewers below. Our responses are in RED.

Thank you again and accept my best regards.

Yours

Wasim T. Al Masoodi

The Corresponding Author

---

## [Editor Report · Decision Letter 1]

12 Aug 2024

Electrolytes as Predictors of Fibro fatigue Scores in Long-COVID Patients

PONE-D-24-09563R1

Dear Dr. Al-Masoodi

We’re pleased to inform you that your manuscript has been judged scientifically suitable for publication and will be formally accepted for publication once it meets all outstanding technical requirements.

Kind regards,

Diana Laila Ramatillah, PhD

Academic Editor

PLOS ONE
---

## [Editor Report · Acceptance letter]

19 Aug 2024

PONE-D-24-09563R1 

PLOS ONE

Dear Dr. Al Masoodi, 

I'm pleased to inform you that your manuscript has been deemed suitable for publication in PLOS ONE. Congratulations! Your manuscript is now being handed over to our production team.

Kind regards, 

on behalf of

Prof Diana Laila Ramatillah 

Academic Editor

PLOS ONE